# NOMAD: Nonlinear Manifold Decoders for Operator Learning

**Jacob H. Seidman** [*]
Graduate Program in Applied Mathematics
and Computational Science
University of Pennsylvania
seidj@sas.upenn.edu

**Georgios Kissas** [*]
Department of Mechanical Engineering
and Applied Mechanics
University of Pennsylvania
gkissas@seas.upenn.edu

**Paris Perdikaris**
Department of Mechanical Engineering
and Applied Mechanics
University of Pennsylvania
pgp@seas.upenn.edu

**George J. Pappas**
Department of Electrical
and Systems Engineering
University of Pennsylvania
pappasg@seas.upenn.edu

## Abstract

Supervised learning in function spaces is an emerging area of machine learning research with applications to the prediction of complex physical systems such as fluid flows, solid mechanics, and climate modeling. By directly learning maps (operators) between infinite dimensional function spaces, these models are able to learn discretization invariant representations of target functions. A common approach is to represent such target functions as linear combinations of basis elements learned from data. However, there are simple scenarios where, even though the target functions form a low dimensional submanifold, a very large number of basis elements is needed for an accurate linear representation. Here we present NOMAD, a novel operator learning framework with a nonlinear decoder map capable of learning finite dimensional representations of nonlinear submanifolds in function spaces. We show this method is able to accurately learn low dimensional representations of solution manifolds to partial differential equations while outperforming linear models of larger size. Additionally, we compare to state-of-the-art operator learning methods on a complex fluid dynamics benchmark and achieve competitive performance with a significantly smaller model size and training cost.

## 1 Introduction

Machine learning techniques have been applied to great success for modeling functions between finite dimensional vector spaces. For example, in computer vision (vectors of pixel values) and natural language processing (vectors of word embeddings) these methods have produced state-of-the-art results in image recognition [18] and translation tasks [48]. However, not all data has an obvious and faithful representation as finite dimensional vectors. In

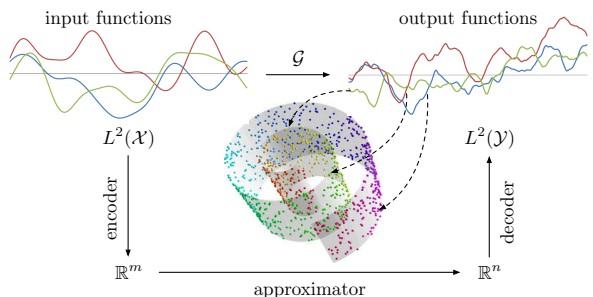

Figure 1: The Operator Learning Manifold Hypothesis.

---

[*]These authors contributed equally.

36th Conference on Neural Information Processing Systems (NeurIPS 2022).

particular, functional data is mathematically represented as a vector in an infinite dimensional vector space. This kind of data appears naturally in problems coming from physics, where scenarios in fluid dynamics, solid mechanics, and kinematics are described by functions of continuous quantities.

Supervised learning in the infinite dimensional setting can be considered for cases where we want to map functional inputs to target functional outputs. For example, we might wish to predict the velocity of a fluid as function of time given an initial velocity field, or predict the pressure field across the surface of the Earth given temperature measurements. This is similar to a finite dimensional regression problem, except that we are now interested in learning an operator between spaces of functions. We refer to this as a *supervised operator learning problem*: given a data-set of $N$ pairs of functions $\{(u^1, s^1), \ldots, (u^N, s^N)\}$, learn an operator $\mathcal{F}$ which maps input functions to output functions such that $\mathcal{F}(u^i) = s^i, \forall i$.

One approach to solve the supervised operator learning problem is to introduce a parameterized operator architecture and train it to minimize a loss between the model's predicted functions and the true target functions in the training set. One of the first operator network architectures was presented in [8] with accompanying universal approximation guarantees in the uniform norm. These results were adapted to deep networks in [30] and led to the DeepONet architecture and its variants [51, 32, 21]. The Neural Operator architecture, motivated by the composition of linear and nonlinear layers in neural networks, was proposed in [27]. Using the Fourier convolution theorem to compute the integral transform in Neural Operators led to the Fourier Neural Operator [28]. Other recent architectures include approaches based on PCA-based representations [2], random feature approaches [36], wavelet approximations to integral transforms [15], and attention-based architectures [23].

A common feature shared among many of these approaches is that they aim to approximate an operator using three maps: an encoder, an approximator, and a decoder, see Figure 1 and Section 3 for more details. In all existing approaches embracing this structure, the decoder is constructed as a linear map. In doing so, the set of target functions is being approximated with a finite dimensional linear subspace in the ambient target function space. Under this setting, the universal approximation theorems of [8, 24, 25] guarantee that there exists a linear subspace of a large enough dimension which approximates the target functions to any prescribed accuracy.

However, as with finite dimensional data, there are scenarios where the target functional data concentrates on a low dimensional *nonlinear* submanifold. We refer to the phenomenon of data in function spaces concentrating on low dimensional submanifolds as the *Operator Learning Manifold Hypothesis*, see Figure 1. For example, it is known that certain classes of parametric partial differential equations admit low dimensional nonlinear manifolds of solution functions [9]. Although linear representations can be guaranteed to approximate these spaces, their required dimension can become very large and thus inefficient in capturing the true low dimensional structure of the data.

In this paper, we are motivated by the Operator Learning Manifold Hypothesis to formulate a new class of operator learning architectures with *nonlinear* decoders. Our key contributions can be summarized as follows.

- **Limitations of Linear Decoders:** We describe in detail the shortcomings of operator learning methods with linear decoders and present some fundamental lower bounds along with an illustrative operator learning problem which is subject to these limitations.

- **Nonlinear Manifold Decoders (NOMAD):** This motivates a novel operator learning framework with a nonlinear decoder that can find low dimensional representations for finite dimensional nonlinear submanifolds in function spaces.

- **Enhanced Dimensionality Reduction:** A collection of numerical experiments involving linear transport and nonlinear wave propagation shows that, by learning nonlinear submanifolds of target functions, we can build models that achieve state-of-the-art accuracy while requiring a significantly smaller number of latent dimensions.

- **Enhanced Computational Efficiency:** As a consequence, the resulting architectures contain a significantly smaller number of trainable parameters and their training cost is greatly reduced compared to competing linear approaches.

We begin our presentation in Section 2 by providing a taxonomy of representative works in the literature. In Section 3 we formally define the supervised operator learning problem and discuss existing approximation strategies, with a focus on highlighting open challenges and limitations. In

Section 4 we present the main contributions of this work and illustrate their utility through the lens of a pedagogical example. In Section 5 we provide a comprehensive collection of experiments that demonstrate the performance of using NOMAD against competing state-of-the-art methods for operator learning. Section 6 summarizes our main findings and discusses lingering limitations and broader impact. Additional details on architectures, hyperparameter selection, and training details are provided in the Supplemental Materials. The code and data accompanying this manuscript are available at https://github.com/PredictiveIntelligenceLab/NOMAD.

## 2 Related Work in Dimensionality Reduction

**Low Dimensional Representations in Finite Dimensional Vector Spaces:** Finding low dimensional representations of high dimensional data has a long history, going back to 1901 with the original formulation of principal components analysis (PCA) [39]. PCA is a linear method that works best when data concentrates on low dimensional subspaces. When data instead concentrates on low dimensional nonlinear spaces, kernelized PCA [42] and manifold learning techniques such as Isomap and diffusion maps [46, 11, 35] can be effective in finding nonlinear low dimensional structure, see [47] for a review. The recent popularity of deep learning has introduced new methods for finding low dimensional structure in high dimensional data-sets, most notably using auto-encoders [52, 6] and deep generative models [12, 22]. Relevant to our work, such techniques have found success in approximating submanifolds in vector spaces corresponding to discretized solutions of parametric partial differential equations (PDEs) [43, 41, 14], where a particular need for nonlinear dimension reduction arises in advection-dominated problems common to fluid mechanics and climate science [26, 33].

**Low Dimensional Representations in Infinite Dimensional Vector Spaces:** The principles behind PCA generalize in a straightforward way to functions residing in low dimensional subspaces of infinite dimensional Hilbert spaces [50]. In the field of reduced order modeling of PDEs this is sometimes referred to as proper orthogonal decomposition [7] (see [29] for an interesting exposition of the discrete version and connections to the Karhunen-Loève decomposition). Affine representations of solution manifolds to parametric PDEs and guarantees on when they are effective using the notion of linear $n$-widths [40] have been explored in [9]. As in the case of finite dimensional data, using a kernel to create a feature representation of a set of functions, and then performing PCA in the associated Reproducing Kernel Hilbert Space can give nonlinear low dimensional representations [45]. The theory behind optimal nonlinear low dimensional representations for sets of functions is still being developed, but there has been work towards defining what "optimal" should mean in this context and how it relates to more familiar geometric quantities [10].

## 3 Operator Learning

**Notation:** Let us first set up some notation and give a formal statement of the supervised operator learning problem. We define $C(\mathcal{X}; \mathbb{R}^d)$ as the set of continuous functions from a set $\mathcal{X}$ to $\mathbb{R}^d$. When $\mathcal{X} \subset \mathbb{R}^n$, we define the Hilbert space,

$$L^2(\mathcal{X}; \mathbb{R}^d) = \left\{ f : \mathcal{X} \to \mathbb{R}^d \mid \|f\|_{L^2}^2 := \int_{\mathcal{X}} \|f(x)\|_{\mathbb{R}^d}^2 \, \mathrm{d}x < \infty \right\}.$$

This is an infinite dimensional vector space equipped with the inner product $\langle f, g \rangle = \int_{\mathcal{X}} f(x)g(x)dx$. When $\mathcal{X}$ is compact, we have that $C(\mathcal{X}; \mathbb{R}^d) \subset L^2(\mathcal{X}; \mathbb{R}^d)$. We now can present a formal statement of the supervised operator learning problem.

**Problem Formulation:** Suppose we are given a training data-set of $N$ pairs of functions $(u^i, s^i)$, where $u^i \in C(\mathcal{X}; \mathbb{R}^{d_u})$ with compact $\mathcal{X} \subset \mathbb{R}^{d_x}$, and $s^i \in C(\mathcal{Y}; \mathbb{R}^{d_s})$ with compact $\mathcal{Y} \subset \mathbb{R}^{d_y}$. Assume there is a ground truth operator $\mathcal{G} : C(\mathcal{X}; \mathbb{R}^{d_u}) \to C(\mathcal{Y}; \mathbb{R}^{d_s})$ such that $\mathcal{G}(u^i) = s^i$ and that the $u^i$ are sampled i.i.d. from a probability measure on $C(\mathcal{X}; \mathbb{R}^{d_u})$. The goal of the *supervised operator learning problem* is to learn a continuous operator $\mathcal{F} : C(\mathcal{X}; \mathbb{R}^{d_x}) \to C(\mathcal{Y}; \mathbb{R}^{d_s})$ to approximate $\mathcal{G}$. To do so, we will attempt to minimize the following empirical risk over a class of

operators $\mathcal{F}_\theta$, with parameters $\theta \in \Theta \subset \mathbb{R}^{d_\theta}$,

$$\mathcal{L}(\theta) := \frac{1}{N} \sum_{i=1}^N \|\mathcal{F}_\theta(u^i) - s^i\|^2_{L^2(\mathcal{Y};\mathbb{R}^{d_u})}. \tag{1}$$

**An Approximation Framework for Operators:** A popular approach to learning an operator $\mathcal{G} : L^2(\mathcal{X}) \to L^2(\mathcal{Y})$ acting on a probability measure $\mu$ on $L^2(\mathcal{X})$ is to construct an approximation out of three maps [25] (see Figure 1),

$$\mathcal{G} \approx \mathcal{F} := \mathcal{D} \circ \mathcal{A} \circ \mathcal{E}. \tag{2}$$

The first map, $\mathcal{E} : L^2(\mathcal{X}) \to \mathbb{R}^m$ is known as the encoder. It takes an input function and maps it to a finite dimensional feature representation. For example, $\mathcal{E}$ could take a continuous function to its point-wise evaluations along a collection of $m$ sensors, or project a function onto $m$ basis functions. The next map $\mathcal{A} : \mathbb{R}^m \to \mathbb{R}^n$ is known as the approximation map. This can be interpreted as a finite dimensional approximation of the action of the operator $\mathcal{G}$. Finally, the image of the approximation map is used to create the output functions in $L^2(\mathcal{Y})$ by means of the decoding map $\mathcal{D} : \mathbb{R}^n \to L^2(\mathcal{Y})$. We will refer to the dimension, $n$, of the domain of the decoder as the *latent dimension*. The composition of these maps can be visualized in the following diagram.

$$
\begin{array}{ccc}
L^2(\mathcal{X}) & \xrightarrow{\ \mathcal{G}\ } & L^2(\mathcal{Y}) \\
{\scriptstyle \mathcal{E}}\downarrow & & \uparrow{\scriptstyle \mathcal{D}} \\
\mathbb{R}^m & \xrightarrow{\ \mathcal{A}\ } & \mathbb{R}^n
\end{array}
\tag{3}
$$

**Linear Decoders:** Many successful operator learning architectures such as the DeepONet [30], the (pseudo-spectral) Fourier Neural Operator in [24], LOCA [23], and the PCA-based method in [2] all use linear decoding maps $\mathcal{D}$. A linear $\mathcal{D}$ can be defined by a set of functions $\tau_i \in L^2(\mathcal{Y})$, $i = 1, \ldots, n$, and acts on a vector $\beta \in \mathbb{R}^n$ as

$$\mathcal{D}(\beta) = \beta_1 \tau_1 + \ldots + \beta_n \tau_n. \tag{4}$$

For example, the functions $\tau_i$ can be built using trigonometric polynomials as in the $\Psi$-FNO [24], be parameterized by a neural network as in DeepONet [30], or created as the normalized output of a kernel integral transform as in LOCA [23].

**Limitations of Linear Decoders:** We can measure the approximation accuracy of the operator $\mathcal{F}$ with two different norms. First is the $L^2(\mu)$ operator norm,

$$\|\mathcal{F} - \mathcal{G}\|^2_{L^2(\mu)} = \underset{u \sim \mu}{\mathbb{E}} \left[ \|\mathcal{F}(u) - \mathcal{G}(u)\|^2_{L^2(\mathcal{Y})} \right]. \tag{5}$$

Note that the empirical risk used to train a model for the supervised operator learning problem (see (1)) is a Monte Carlo approximation of the above population loss. The other option to measure the approximation accuracy is the uniform operator norm. For a compact set $\mathcal{U} \subset L^2(\mathcal{X})$ we can define the error

$$\sup_{u \in \mathcal{U}} \|\mathcal{F}(u) - \mathcal{G}(u)\|_{L^2(\mathcal{Y})}. \tag{6}$$

When a linear decoder is used for $\mathcal{F} = \mathcal{D} \circ \mathcal{A} \circ \mathcal{E}$, a data-dependent lower bound to each of these errors can be derived.

$L^2$ **lower bound:** When the pushforward measure has a finite second moment, its covariance operator $\Gamma : L^2(\mathcal{Y}) \to L^2(\mathcal{Y})$ is self-adjoint, positive semi-definite, and trace-class, and thus admits an orthogonal set of eigenfunctions spanning its image, $\{\phi_1, \phi_2, \ldots\}$ with associated decreasing eigenvalues $\lambda_1 \geq \lambda_2 \geq \ldots$. The decay of these eigenvalues indicates the extent to which samples from $\mathcal{G}_{\#}\mu$ concentrate along the leading finite-dimensional eigenspaces. It was shown in [25] that for any choice of $\mathcal{E}$ and $\mathcal{A}$, these eigenvalues give a fundamental lower bound to the expected squared $L^2$ error of the operator learning problem with architectures as in (3) using a linear decoder $\mathcal{D}$,

$$\underset{u \sim \mu}{\mathbb{E}} \left[ \|\mathcal{D} \circ \mathcal{A} \circ \mathcal{E}(u) - \mathcal{G}(u)\|^2_{L^2} \right] \geq \sum_{k > n} \lambda_k. \tag{7}$$

This result can be refined to show that the optimal choice of functions $\tau_i$ (see equation (4)) for a linear decoder are the leading $n$ eigenfunctions of the covariance operator $\{\phi_1, \ldots, \phi_n\}$. The interpretation of this result is that the best way to approximate samples from $\mathcal{G}_\# \mu$ with an $n$-dimensional subspace is to use the subspace spanned by the first $n$ "principal components" of the probability measure $\mathcal{G}_\# \mu$. The error incurred by using this subspace is determined by the remaining principal components, namely the sum of their eigenvalues $\sum_{k>n} \lambda_k$. The operator learning literature has noted that for problems with a slowly decaying pushforward covariance spectrum (such as solutions to advection-dominated PDEs) these lower bounds cause poor performance for models of the form (3) [25, 13].

**Uniform lower bound:**  In the reduced order modelling of PDEs literature [9, 10, 26] there exists a related notion for measuring the degree to an $n$-dimensional subspace can approximate a set of functions $\mathcal{S} \subset L^2(\mathcal{Y})$. This is known as the Kolmogorov $n$-width [40], and for a compact set $\mathcal{S}$ is defined as

$$d_n(\mathcal{S}) = \inf_{\substack{V_n \subset L^2(\mathcal{Y}) \\ V_n \text{ is a subspace} \\ \dim(V_n) = n}} \sup_{s \in \mathcal{S}} \inf_{v \in V_n} \|s - v\|_{L^2(\mathcal{Y})}. \tag{8}$$

This measure of how well a set of functions can be approximated by a linear subspace in the uniform norm leads naturally to a lower bound for the uniform error (6). To see this, first note that for any $u \in \mathcal{U}$, the error from $\mathcal{F}(u)$ to $\mathcal{G}(u)$ is bounded by the minimum distance from $\mathcal{G}(u)$ to the image of $\mathcal{F}$. For a linear decoder $\mathcal{D} : \mathbb{R}^n \to L^2(\mathcal{Y})$, define the (at most) $n$-dimensional $V_n = \text{im}(\mathcal{D}) \subset L^2(\mathcal{Y})$. Note that $\text{im}(\mathcal{F}) \subseteq V_n$, and we may write

$$\|\mathcal{F}(u) - \mathcal{G}(u)\|_{L^2(\mathcal{Y})} \geq \inf_{v \in V_n} \|v - \mathcal{G}(u)\|_{L^2(\mathcal{Y})}.$$

Taking the supremum of both sides over $u \in \mathcal{U}$, and then the infimum of both sides over all $n$-dimensional subspaces $V_n$ gives

$$\sup_{u \in \mathcal{U}} \|\mathcal{F}(u) - \mathcal{G}(u)\|_{L^2(\mathcal{Y})} \geq \inf_{\substack{V_n \subset L^2(\mathcal{Y}) \\ V_n \text{ is a subspace} \\ \dim(V_n) = n}} \sup_{u \in \mathcal{U}} \inf_{v \in V_n} \|v - \mathcal{G}(u)\|_{L^2(\mathcal{Y})}.$$

The quantity on the right is exactly the Kolmogorov $n$-width of $\mathcal{G}(\mathcal{U})$. We have thus proved the following complementary statement to (7) when the error is measured in the uniform norm.

**Proposition 1** *Let $\mathcal{U} \in L^2(\mathcal{X})$ be compact and consider an operator learning architecture as in (3), where $\mathcal{D} : \mathbb{R}^n \to L^2(\mathcal{Y})$ is a linear decoder. Then, for any $\mathcal{E} : L^2(\mathcal{X}) \to \mathbb{R}^p$ and $\mathcal{A} : \mathbb{R}^p \to \mathbb{R}^n$, the uniform norm error of $\mathcal{F} := \mathcal{D} \circ \mathcal{A} \circ \mathcal{E}$ satisfies the lower bound*

$$\sup_{u \in \mathcal{U}} \|\mathcal{F}(u) - \mathcal{G}(u)\|_{L^2(\mathcal{Y})} \geq d_n(\mathcal{G}(\mathcal{U})). \tag{9}$$

Therefore, we see that in both the $L^2(\mu)$ and uniform norm, the error for an operator learning problem with a linear decoder is fundamentally limited by the extent to which the space of output functions "fits" inside a finite dimensional linear subspace. In the next section we will alleviate this fundamental restriction by allowing decoders that can learn nonlinear embeddings of $\mathbb{R}^n$ into $L^2(\mathcal{Y})$.

## 4   Nonlinear Decoders for Operator Learning

**A Motivating Example:**  Consider the problem of learning the antiderivative operator mapping functions to their first-order derivative

$$\mathcal{G} : u \mapsto s(x) := \int_0^x u(y) \, dy, \tag{10}$$

acting on a set of input functions

$$\mathcal{U} := \left\{ u(x) = 2\pi t \cos(2\pi t x) \, \big| \, 0 \leq t_0 \leq t \leq T \right\}. \tag{11}$$

The set of output functions is given by $\mathcal{G}(\mathcal{U}) = \{\sin(2\pi t x) \mid 0 < t_0 < t < T\}$. This is a one-dimensional curve of functions in $L^2([0, 1])$ parameterized by a single number $t$. However, we would not be able to represent this set of functions with a one-dimensional linear subspace. In Figure 2b

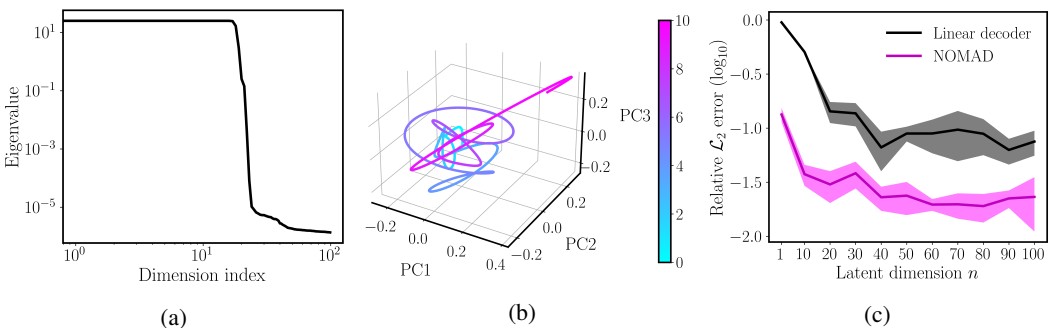

(a)  (b)  (c)

Figure 2: *Antiderivative Example:* (a) log of the leading 100 PCA eigenvalues of $\mathcal{G}(\mathcal{U})$; (b) Projection of functions in the image of $\mathcal{G}(\mathcal{U})$ on the first three PCA components, colored by the frequency of each projected function; (c) Relative $L^2$ testing error ($\log_{10}$ scale) as a function of latent dimension $n$ for linear and nonlinear decoders (over 10 independent trials).

we perform PCA on the functions in this set evaluated on a uniform grid of values of $t$. We see that the first 20 eigenvalues are nonzero and relatively constant, suggesting that an operator learning architecture with a linear or affine decoder would need a latent dimension of at least 20 to effectively approximate functions from $\mathcal{G}(\mathcal{U})$. Figure 2a gives a visualization of this curve of functions projected onto the first three PCA components. We will return to this example in Section 5, and see that an architecture with a nonlinear decoder can in fact approximate the target output functions with superior accuracy compared to the linear case, using a single latent dimension that can capture the underlying nonlinear manifold structure.

**Operator Learning Manifold Hypothesis:**  We now describe an assumption under which a non-linear decoder is expected to be effective, and use this to formulate the NOMAD architecture. To this end, let $\mu$ be a probability measure on $L^2(\mathcal{X})$ and $\mathcal{G} : L^2(\mathcal{X}) \to L^2(\mathcal{Y})$. We assume that there exists an $n$-dimensional manifold $\mathcal{M} \subseteq L^2(\mathcal{Y})$ and an open subset $\mathcal{O} \subset \mathcal{M}$ such that

$$\mathbb{E}_{u\sim\mu}\left[\inf_{v\in\mathcal{O}}\|\mathcal{G}(u)-v\|_{L^2}^2\right]\leq\epsilon. \tag{12}$$

In connection with the manifold hypothesis in deep learning [5, 4], we refer to this as the *Operator Learning Manifold Hypothesis*. There are scenarios where it is known this assumption holds, such as in learning solutions to parametric PDEs [33].

This assumption motivates the construction of a nonlinear decoder for the architecture in (3) as follows. For each $u$, choose $v(u) \in \mathcal{O}$ such that

$$\mathbb{E}_{u\sim\mu}\left[\|\mathcal{G}(u)-v(u)\|_{L^2}^2\right]\leq\epsilon. \tag{13}$$

Let $\phi : \mathcal{O} \to \mathbb{R}^n$ be a coordinate chart for $\mathcal{O} \subset \mathcal{M}$. We can represent $v(u) \in \mathcal{O}$ by its coordinates $\phi(v(u)) \in \mathbb{R}^n$. Consider a choice of encoding and approximation maps such that $\mathcal{A}(\mathcal{E}(u))$ gives the coordinates for $v(u)$. If the decoder were chosen as $\mathcal{D} := \phi^{-1}$ then by construction, the operator $\mathcal{F} := \mathcal{D} \circ \mathcal{A} \circ \mathcal{E}$ will satisfy

$$\mathbb{E}_{u\sim\mu}\left[\|\mathcal{G}(u)-\mathcal{F}(u)\|_{L^2}^2\right]\leq\epsilon. \tag{14}$$

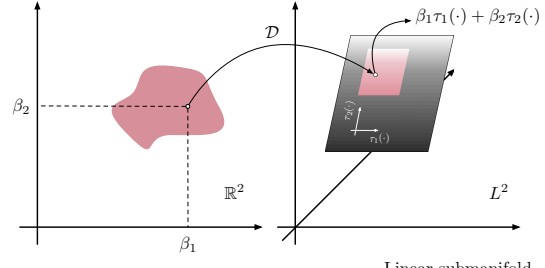

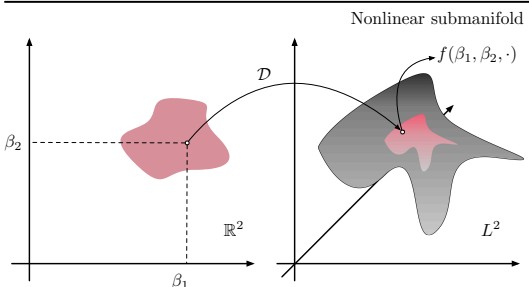

Figure 3: An example of linear versus nonlinear decoders.

Therefore, we interpret a learned decoding map as attempting to give a finite dimensional coordinate system for the solution manifold. Consider a generalized decoder of the following form

$$\tilde{\mathcal{D}} : \mathbb{R}^n \times \mathcal{Y} \to \mathbb{R}. \tag{15}$$

This induces a map from $\mathcal{D} : \mathbb{R}^n \to L^2(\mathcal{Y})$, as $\mathcal{D}(\beta) = \tilde{\mathcal{D}}(\beta, \cdot)$. If the solution manifold $\mathcal{M}$ is a finite dimensional linear subspace in $L^2(\mathcal{Y})$ spanned by $\{\tau_i\}_{i=1}^n$, we would want a decoder to use the coefficients along the basis as a coordinate system for $\mathcal{M}$. A generalized decoder could learn this basis as the output of a deep neural network to act as

$$\tilde{\mathcal{D}}_{\text{lin}}(\beta, y) = \beta_1 \tau_1(y) + \ldots + \beta_n \tau_n(y). \tag{16}$$

However, if the solution manifold is not linear, then we should learn a nonlinear coordinate system given by a nonlinear $\mathcal{D}$. A nonlinear version of $\tilde{\mathcal{D}}$ can be parameterized by using a deep neural network $f : \mathbb{R}^n \times \mathcal{Y} \to \mathbb{R}$ which jointly takes as arguments $(\beta, y)$,

$$\tilde{\mathcal{D}}(\beta, y) = f(\beta, y). \tag{17}$$

When used in the context of an operator learning architecture of the form (3), we call a nonlinear decoder from (17) NOMAD (NOnlinear MAnifold Decoder). Figure 3 presents a visual comparison between linear and nonlinear decoders.

**Summary of NOMAD:**   Under the assumption of the *Operator Learning Manifold Hypothesis*, we have proposed a fully nonlinear decoder (17) to represent target functions using architectures of the form (3). We next show that using a decoder of the form (17) results in operator learning architectures which can learn nonlinear low dimensional solution manifolds. Additionally, we will see that when these solution manifolds do not "fit" inside low dimensional linear subspaces, architectures with linear decoders will either fail or require a significantly larger number of latent dimensions.

## 5   Results

In this section we investigate the effect of using a linear versus nonlinear decoders as building blocks of operator learning architecture taking the form (3). In all cases, we will use an encoder $\mathcal{E}$ which takes point-wise evaluations of the input functions, and an approximator map $\mathcal{A}$ given by a deep neural network. The linear decoder parametrizes a set of basis functions that are learned as the outputs of an MLP network. In this case, the resulting architecture exactly corresponds to the DeepONet model from [30]. We will compare this against using NOMAD where the nonlinear decoder is built using an MLP network that takes as inputs the concatenation of $\beta \in \mathbb{R}^n$ and a given query point $y \in \mathcal{Y}$. All models are trained with by performing stochastic gradient descent on the loss function in (1). The reported errors are measured in the relative $L^2(\mathcal{Y})$ norm by averaging over all functional pairs in the testing data-set. More details about architectures, hyperparameters settings, and training details are provided in the Supplemental Materials.

**Learning the Antiderivative Operator:**   First, we revisit the motivating example from Section 4, where the goal is to learn the antiderivative operator (10) acting on the set of functions (11). In Figure 2c we see the performance of a model with a linear decoder and NOMAD over a range of latent dimensions $n$. For each choice of $n$, 10 experiments with random initialization seeds were performed, and the mean and standard deviation of testing errors are reported. We see that the NOMAD architecture consistently outperforms the linear one (by one order of magnitude), and can even achieve a 10% relative prediction error using only $n = 1$.

**Solution Operator of a Parametric Advection PDE:**   Here we consider the problem of learning the solution operator to a PDE describing the transport of a scalar field with conserved energy,

$$\frac{\partial}{\partial t} s(x, t) + \frac{\partial}{\partial x} s(x, t) = 0, \tag{18}$$

over a domain $(x, t) \in [0, 2] \times [0, 1]$. The solution operator maps an initial condition $s(x, 0) = u(x)$ to the solution at all times $s(x, t)$ which satisfies (18). We consider a training data-set of initial conditions taking the form of radial basis functions with a very small fixed lengthscale centered at randomly chosen locations in the interval $[0, 1]$. We create the output functions by evolving these initial conditions forward in time for 1 time unit according to the advection equation (18) (see Supplemental Materials for more details). Figure 4a gives an illustration of one such solution plotted over the space-time domain.

Performing PCA on the solution functions generated by these initial conditions shows a very slow decay of eigenvalues (see Figure 4b), suggesting that methods with linear decoders will require

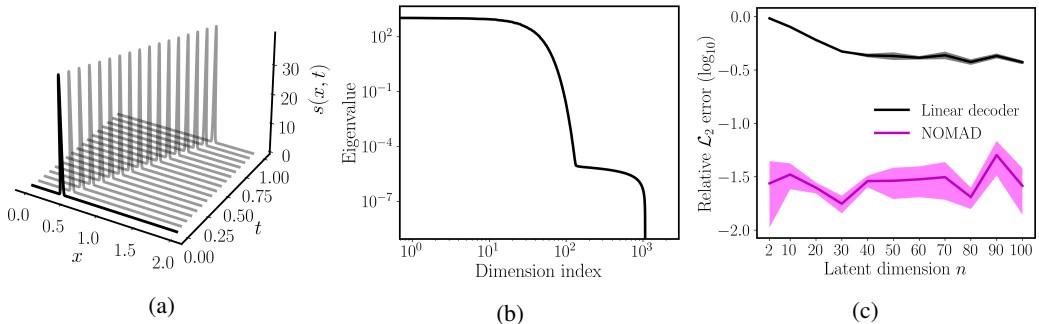

(a)                 (b)                 (c)

Figure 4: *Advection Equation:* (a) Propagation of an initial condition function (highlighted in black) through time according to (18); (b) log of the leading $1,000$ PCA eigenvalues of $\mathcal{G}(\mathcal{U})$; (c) Relative $L^2$ testing error ($\log_{10}$ scale) as a function of latent dimension $n$ for linear and nonlinear decoders (over 10 independent trials).

a moderately large number of latent dimensions. However, since the data-set was constructed by evolving a set of functions with a single degree of freedom (the center of the initial conditions), we would expect the output functions to form a solution manifold of very low dimension.

In Figure 4c we compare the performance of a linear decoder and NOMAD as a function of the latent dimension $n$. Linear decoders yield poor performance for small values of $n$, while NOMAD appears to immediately discover a good approximation to the true solution manifold.

**Propagation of Free-surface Waves:**   As a more challenging benchmark we consider the shallow-water equations; a set of hyperbolic equations that describe the flow below a pressure surface in a fluid [49]. The underlying PDE system takes the form

$$\frac{\partial \boldsymbol{U}}{\partial t} + \frac{\partial \boldsymbol{F}}{\partial x} + \frac{\partial \boldsymbol{G}}{\partial y} = 0, \quad \boldsymbol{U} = \begin{pmatrix} \rho \\ \rho v_1 \\ \rho v_2 \end{pmatrix}, \quad \boldsymbol{F} = \begin{pmatrix} \rho v_1 \\ \rho v_1^2 + \frac{1}{2} g \rho^2 \\ \rho v_1 v_2 \end{pmatrix}, \quad \boldsymbol{G} = \begin{pmatrix} \rho v_2 \\ \rho v_1 v_2 \\ \rho v_2^2 + \frac{1}{2} g \rho^2 \end{pmatrix}, \quad (19)$$

where $\rho(x, y, t)$ the fluid height from the free surface, $g$ is the gravity acceleration, and $v_1(x, y, t)$, $v_2(x, y, t)$ denote the horizontal and vertical fluid velocities, respectively. We consider reflective boundary conditions and random initial conditions corresponding to a random droplet falling into a still fluid bed (see Supplemental Materials). In Figure 5a we show the average testing error of a model with a linear and nonlinear decoder as a function of the latent dimension. Figure 5b shows snapshots of the predicted surface height function on top of a plot of the errors to the ground truth for the best, worst, median, and a random sample from the testing data-set.

We additionally use this example to compare the performance of a model with a linear decoder and NOMAD to other state-of-the-art operator learning architectures. For this experiment we include a harmonic feature expansion in the DeepONet, as it was shown in [32] that this can significantly improve its performance. We do not include this in the NOMAD architecture for this experiment (see Supplemental Materials for more details). In Table 1 we present the mean relative error and its standard deviation for different operator learning methods, as well as the prediction that provides the worst error in the testing data-set when compared against the ground truth solution. For each method we also report the number of its trainable parameters, the number of its latent dimension $n$, and the training wall-clock time in minutes. Since the general form of the FNO [28] does not neatly fit into the architecture given by (3), there is not a directly comparable measure of latent dimension for it. We also observe that, although the model with NOMAD closely matches the performance of LOCA [23], its required latent dimension, total number of trainable parameters, and total training time are all significantly smaller.

## 6   Discussion

**Summary:**   We have presented a novel framework for supervised learning in function spaces. The proposed methods aim to address challenging scenarios where the manifold of target functions has

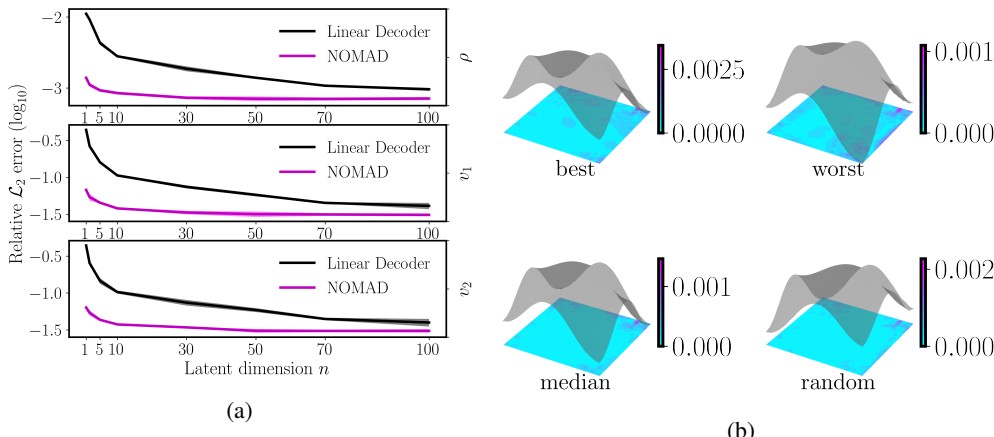

(a)

(b)

Figure 5: *Propagation of Free-surface Waves:* (a) Relative $L^2$ testing error ($\log_{10}$ scale) as a function of latent dimension $n$ for linear and nonlinear decoders (over 10 independent trials); (b) Visualization of predicted free surface height $\rho(x, y, t = 0.31)$ and point-wise absolute prediction error contours corresponding to the best, worst, and median samples in the test data-set, along with a representative test sample chosen at random.

Table 1: Comparison of relative $L^2$ errors (in %) for the predicted output functions for the shallow water equations benchmark against existing state-of-the-art operator learning methods: LOCA [23], DeepONet (DON) [30], and the Fourier Neural Operator (FNO) [28]. The fourth column reports the relative $L^2$ error for $(\rho, v_1, v_2)$ corresponding to the worst case example in the test data-set. Also shown is each model's total number of trainable parameters $d_\theta$, latent dimension $n$, and computational cost in terms of training time (minutes).

| Method | $\rho$ | $v_1$ | $v_2$ | worst case | $d_\theta$ | $n$ | cost |
|--------|--------|-------|-------|------------|------------|-----|------|
| LOCA | $\mathbf{0.040 \pm 0.015}$ | $2.7 \pm 0.3$ | $2.9 \pm 0.4$ | $(\mathbf{0.1}, \mathbf{3.5}, \mathbf{4.2})$ | $O(10^6)$ | 480 | 12.1 |
| DON | $0.100 \pm 0.030$ | $5.5 \pm 1.2$ | $5.9 \pm 1.4$ | $(0.6, 11, 11)$ | $O(10^6)$ | 480 | 15.4 |
| FNO | $0.140 \pm 0.060$ | $3.4 \pm 1.2$ | $3.5 \pm 1.2$ | $(0.4, 8.9, 8.7)$ | $O(10^6)$ | N/A | 14.0 |
| NOMAD | $0.048 \pm 0.017$ | $\mathbf{2.0 \pm 0.4}$ | $\mathbf{2.6 \pm 0.3}$ | $(0.1, 5.8, 4.9)$ | $\mathbf{O(10^5)}$ | $\mathbf{20}$ | $\mathbf{5.5}$ |

low dimensional structure, but is embedded nonlinearly into its associated function space. Such cases commonly arise across diverse functional observables in the physical and engineering sciences (e.g. turbulent fluid flows, plasma physics, chemical reactions), and pose a significant challenge to the application of most existing operator learning methods that rely on linear decoding maps, forcing them to require an excessively large number of latent dimensions to accurately represent target functions. To address this shortcoming we proposed a fully nonlinear framework that can effectively learn low dimensional representations of nonlinear embeddings in function spaces, and demonstrated that it achieves competitive accuracy to state-of-the-art operator learning methods while using a significantly smaller number of latent dimensions, leading to lighter model parametrizations and reduced training cost.

**Limitations:** Our proposed approach relies on the *Operator Learning Manifold Hypothesis* (see equation (12)), suggesting that cases where a low dimensional manifold structure does not exist will be hard to tackle (e.g. target function manifolds with fractal structure, solutions to evolution equations with strange attractors). Moreover, even when the manifold hypothesis holds, the underlying effective latent embedding dimension is typically not known a-priori, and may only be precisely found via cross-validation. Another direct consequence of replacing linear decoders with fully nonlinear maps is that the lower bound in (9) needs to be rephrased in terms of a nonlinear $n$-width, which in general can be difficult to quantify. Finally, in this work we restricted ourselves to exploring simple nonlinear decoder architectures such as an MLPs with the latent parameters $\beta$ and query location $y$ concatenated as inputs. Further investigation is needed to quantify the improvements that could be brought by considering more contemporary deep learning architectures, such as hypernetworks [16] which can

define input dependent weights for complicated decoder architectures. One example of this idea in the context of reduced order modeling can be found in Pan *et. al.* [37], where the authors propose a hypernetwork based method combined with a Implicit Neural Representation network [44].

**Potential Negative Societal Impacts:** Our contributions enables advances in accelerating the modeling, simulation, and optimization of physical and engineering systems. As with any tool that furthers our understanding and ability to predict the outcomes of complex systems, there may be ill-intentioned use cases, but we do not expect any specific negative impact from this work.

## Acknowledgments and Disclosure of Funding

G.K. and P.P. would like to acknowledge support from the US Department of Energy under under the Advanced Scientific Computing Research program (grant DE-SC0019116), the US Air Force (grant AFOSR FA9550-20-1-0060), and US Department of Energy/Advanced Research Projects Agency (grant DE-AR0001201). J.S. and G.P. would like to acknowledge support from the AFOSR under grant FA9550-19-1-0265 (Assured Autonomy in Contested Environments) and the NSF Simmons Mathematical and Scientific Foundations of Deep Learning (grant 2031985). We also thank the developers of the software that enabled our research, including JAX [3], Kymatio [1], Matplotlib [20], Pytorch [38] and NumPy [17].

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
