# OpenReview forum: "NOMAD: Nonlinear Manifold Decoders for Operator Learning"
_NeurIPS.cc/2022/Conference — NeurIPS 2022 Accept_

### Official Review · Reviewer_uiyi · 2022-07-11

**Rating:** 6
**Confidence:** 4
**Soundness:** 3 good
**Presentation:** 3 good
**Contribution:** 3 good

**Summary:**

The paper proposes a nonlinear decoder in a computational framework for supervised operator learning, which maps a function space to another function space. The paper has presented a general interpretation of operator learning, which consists of encoder-approximation map-decoder and proposes to use a nonlinear decoder instead of existing linear decoders.

**Questions:**

- how does the size of the latent dimension affect the performance? In Fig 4.C, increasing the latent dimension has no effect or even increases the l2 error.
- would there be any general suggestion or guidance in choosing the size of latent dimension/ neural network choices ?
- in the basis point of view, how does the dimension of the last to the second layer affect the performance? if the last layer is the linear layer, the columns of the weight matrix in the last layer can be considered as a basis. How does the number of the columns in the last layer affect the performance?

**Limitations:**

the authors adequately addressed the limitations in the manuscript.

**Strengths And Weaknesses:**

[+] well-written and well-motivated

[+] proposes a simple, but essential extension to existing the operator learning paradigm

[-] hyperparameter search to some extent would make the contribution stronger and convincing. In the current presentation, several things are unclear (in particular, deep learning aspects) e.g., how the size of latent dimension affects the performance, how the performance would be if the decoder is more powerful, if a more expressive decoder is employed, even lower dimensional latent space can be employed? would a specific nonlinearity or learning rate result in better performance and so on. Some intuitions or elaborations on rationale behind those choices would be appreciated.

[-] although the authors add references for neural network architectures used in the experiments, those information could be included in the manuscript directly.

---

> ### Author Response · Authors · 2022-08-02
> **Response to Reviewer uiyi**
>
> We thank the reviewer for their time and feedback.
>
> As with all deep learning methods, an extensive search over hyperparameters can give insight into the best possible performance for a given problem scenario guidelines on how to make hyperparameter choices are useful for practitioners applying these methods.  Our goal in this work was not to make a statement of all model settings which result in the highest possible performance, but to highlight the fundamental need for nonlinear decoder architectures in operator learning problems.  For a fixed latent dimension, the lower bound associated with a linear decoder is independent of any additional architecture choices such as its hidden widths, depth, or activation functions, as well as training hyperparameters such as learning rates.  A NOMAD architecture can be implemented with different choices for all of these components and the reviewer is correct in identifying that a search over these choices as an important area of future research.
>
> We thank the reviewer for the suggestion to include more architecture details in the manuscript instead of having them  as a citation.  These have been added to the Supplemental Material.
>
> Increasing the latent dimension will generally increase model performance, though increasing this also results in larger neural network architectures that could take slightly longer to train.  We believe this is the cause of the results presented in Figure 4.C.  There are some scenarios where a good estimate of a sufficient latent dimension can be guessed a priori.  For example when solving parametric PDE problems with a known number of degrees of freedom in the input functions, such as the antiderivative and advection example in the paper, this number can be taken as a starting point to find a good latent dimension.
>
> As we note in the limitations section, at the moment the most reliable way to choose a latent dimension is through cross-validation and a sweep over the possible values.  Other approaches could take the form of first performing a nonlinear dimensionality reduction scheme on the set of output functions, and then using this as a starting point for a latent dimension.  While there exist several interesting approaches for determining the intrinsic (potentially nonlinear) dimension of finite dimensional datasets, for example
>
> Levina, Elizaveta, and Peter Bickel. "Maximum likelihood estimation of intrinsic dimension." Advances in neural information processing systems 17 (2004),
>
> many of these approaches have not yet been extended to functional data.  This is a very interesting direction for future research with immediate applications to NOMAD.
>
> Guidelines for choosing architectures of the encoding and approximating map will be generally be problem-specific.  For example, input functions over a two dimensional spatial domain with spatial correlations would suggest using a convolutional architecture to map the input into the latent space. Similarly, when input functions are defined over periodic domains, encoding with Fourier modes can be quite effective.  On the decoder side, we intentionally used the simplest possible version of a nonlinear decoder (a simple fully connected MLP) to investigate the effect of a nonlinear map in isolation of any other modeling choices and without tapping into any additional problem structure.
>
> To address the last question, we would like to clarify that the basis point of view presented in the paper is a basis in the space of output functions, not the codomain of these functions.  For ease of explanation, let the output functions be scalar valued.  In DeepONet, the decoder is constructed from a neural network which only takes as inputs the output function argument $y$.  The outputs of this network have the dimension of the latent space, $n$.  This network as defines $n$ functions of $y$ which we interpret as a basis for the subspace of functions the DeepONet is able to represent. Therefore, the number of columns in the weight matrix of the last layer of this network correspond to the latent dimension (and therefore number of basis vectors for the subspace of output functions) but the columns of this matrix alone do not define this basis.

---

> > ### Comment · Reviewer_uiyi · 2022-08-08
> > **Response to the authors**
> >
> > Thank you for providing responses. Many of my concerns have been addressed by the authors’ response, in particular, the clarification on the basis point of view. Although there seem to be other concerns to be resolved, e.g., generalizability of the proposed idea to other complex problems, the second reading of this paper seems to provide more merit than the first reading. Thus, I’ve updated my rating accordingly.

---

### Official Review · Reviewer_MKwA · 2022-07-12

**Rating:** 6
**Confidence:** 4
**Soundness:** 3 good
**Presentation:** 3 good
**Contribution:** 2 fair

**Summary:**

This paper deals with supervised regression when both the input and output data are functions, with application to partial differential equations (PDEs). Classical techniques include a linear modelling of the relationship between the input and output variables, which can then take the form Linear Encoder -> Linear Operator -> Linear Decoder.

In this work, authors show the theoretical limitations of taking the decoder to be a linear map, and propose a novel neural architecture based on a nonlinear decoder that allows to learn operators with a smaller latent dimension in the "operator learning manifold hypothesis" scenario. Experimental results illustrate the benefit of the approach on several benchmarks corresponding to various PDEs.

**Questions:**

Questions:

- Despite the PDEs being linear, the mapping to learn may no be linear, so why is the linear modelling necessary ? Especially, one could use vector-valued RKHSs to model such relationship, with the additional benefit of being able to regularize the empirical risk using the vv-RKHS norm.
- Concerning the $f(\beta, y)$ network, since $y$ is a function does it need to go through the encoder before entering $f$ ? Could you precise the architecture of such a network ?

Suggestions:
- Line 227 could be clarified: if $\tilde{D}$ is a generalized decoder, then $D$ does not have to be linear in $y$.

**Limitations:**

The authors have adequately addressed the limitations and potential negative societal impact of their work.

**Strengths And Weaknesses:**

Strengths:

- The paper is clear and easy to follow.
- References are well organized.
- Topic of operator learning is relevant to the NeurIPS community and is likely to have multi-disciplinary benefits.
- Experimental section is convincing.

Weaknesses:

- The technical contribution of the paper appears moderate, as it mainly consists in replacing the linear decoder by a nonlinear one obtained with a neural network. The part about the limitations of the linear decoders reads more as a remark than a real theoretical contribution.
- There are some levels of approximation which are not discussed in the paper. Especially, by adopting the functional space view, the fact that one cannot directly build a neural net $f(\beta, y)$ when $y$ is a function is swept under the rug.

---

> ### Author Response · Authors · 2022-08-02
> **Response to Reviewer MKwA**
>
> We thank the reviewer for taking the time to read our paper and their feedback.
>
> We first would like to clarify that the presence of a linear decoder does not necessarily mean that the overall operator architecture is linear as well.  As described in Section 3, the architectures we consider in this paper are compositions of the encoding, approximation, and decoding maps.  In all models presented, including NOMAD, the approximation map is nonlinear (typically a neural network).  This makes the overall operator nonlinear as a map between the input and output function spaces.  To summarize, NOMAD and the other mentioned operator learning approaches all learn a nonlinear operator between function spaces as a composition of maps, even though some of these composition building blocks may be linear.
>
> What we show in this paper is that even with a nonlinear operator architecture, the presence of a linear decoder leads to some fundamental limitations that can be addressed by replacing it with a nonlinear decoder.  This limitation has only been addressed indirectly in the literature, see
>
> Samuel Lanthaler, Siddhartha Mishra, and George E Karniadakis. Error estimates for Deep-
> ONets: A deep learning framework in infinite dimensions. Transactions of Mathematics and Its
> Applications, 6(1):tnac001, 2022
>
> and its solution via nonlinear decoders has not been explored. Therefore, we feel the discussion on linear decoders is an important contribution to the operator learning community in order to build intuition about how these methods can fail and motivate the clear need for nonlinear decoder architectures. Considering non-linear decoders is not limited to the architecture presented in this paper and therefore can have a broad impact to designing operator learning methods.
>
> Lastly, we would like to clarify that $y$ is a point in the domain of the output function, not a function itself.  Therefore, a neural network which takes as inputs the concatenation of $\beta$ and $y$ and outputs a point in the codomain of the output function is well defined.  As mentioned in the paper, there are other architectural choices that can be made to combine $\beta$ and $y$ for producing outputs that depend nonlinearly on the $\beta$'s.  This is an important area for future work that is likely to yield improved results.

---

> > ### Comment · Reviewer_MKwA · 2022-08-07
> > **Response to authors**
> >
> > I would like to thank the authors for the constructive feedback.
> >
> > I understand better the part about $y$ being a point in the domain of the output function, my criticism on this was unsubstantiated.
> >
> > I still think that the technical contribution of the paper is quite light, thus keeping my score as it is.

---

### Official Review · Reviewer_Dpah · 2022-07-14

**Rating:** 5
**Confidence:** 4
**Soundness:** 1 poor
**Presentation:** 2 fair
**Contribution:** 1 poor

**Summary:**

The authors consider a functional regression problem, i.e., given a dataset of pairs of functions, learn an operator which maps input functions to output functions. They represent the operator as a combination of an encoder, an approximator in a latent space, and a decoder that maps a latent representation into a target function. The authors claimed that a typical decoder is linear, i.e., it outputs a linear combination of some basis functions, which depends on the input vector of the predicted target function. However, in the case of a nonlinear manifold, we should use a nonlinear decoder, so the authors proposed to use a nonlinear decoder represented by a neural network. They demonstrated the usefulness of such modification based on several examples, including cases of PDE systems, which maps some functional input condition into an output functional solution.

**Questions:**

- 79: The authors mentioned “EnhancedComputationalEfficiency”. However, they did not provide any information about the computational efficiency in the text of the paper.
- 132: here, mu is a probability measure on L^2(X). (5): here, mu is a probability measure on an infinite-dimensional space of output functions. There is some contradiction in notations. What is U in (5)? It is not defined earlier in the text.
- 260: antidervative -> antidirevative?

**Limitations:**

- There is no ablation study. The authors did not provide any results on how accuracy depends on various hyperparameters, sampling rates of the input/output functions, etc.
- There is no comparison with SOTA functional regression approaches.
- If the input/output functions are complex, we should use some very dense grid to represent them. However, in such case, it is not clear which architectural choices we should make about the structure of the neural networks based encoder/decoder/approximator

**Strengths And Weaknesses:**

Strength
- The problem of functional regression is important for applications. So the topic of the paper is important
- The authors provided some simple theoretical analysis; in particular, they interpreted the approximation error as the Kolmogorov n-width and demonstrated that in the case of a linear decoder, the approximation error is lower bounded by the n-width. I consider this a trivial result; moreover, the authors did not use it anyhow in their algorithm; however, the interpretation can be useful.
- The authors demonstrated that the nonlinear decoder provides a smaller approximation error

Weakness
- The novelty of the paper is limited. The main improvement is that the authors used a nonlinear decoder represented by a neural network.
- The experimental section is very limited. It does not contain any comparison with various competing methods. For example, various neural networks based realizations of the so-called Koopman operator (https://www.ias.informatik.tu-darmstadt.de/uploads/Team/JoeWatson/Damken_Bsc_2020.pdf) can also be used for solving the same type of a problem. Also, what about comparing the considered approach to the approach from https://arxiv.org/abs/1506.07365 , which can be easily adapted to the considered case? A solution of a PDE can be represented as a tensor. What about comparing the proposed approach to SOTA image2image approaches based on deep neural networks?

---

> ### Author Response · Authors · 2022-08-02
> **Response to Reviewer Dpah**
>
> We thank the reviewer for their time and feedback about the paper.
>
> First, we would like to address the reviewers concerns about the comparisons to competing methods.  We respectfully disagree that a comparison to a Koopman based approach or the E2C method from https://arxiv.org/abs/1506.07365 is relevant for the current paper.  The problem addressed in both of those papers is to find an alternate state space representation of a dynamical system (whether through Koopman observable or a low dimensional latent representation of the state of the dynamical system), so that the dynamics are more easily identified and controlled through the latent state space.  If a functional regression problem is defined as in our paper, where we are given a dataset of input functions $u^i$ and aim to find an operator that maps these to a known set of output functions $s^i$, it is not clear how these two methods are general functional regression approaches.
>
> The comparison of operator learning methods to image-to-image approaches has been extensively discussed previously in the literature, see
>
> Li, Zongyi, et al. "Neural operator: Graph kernel network for partial differential equations." arXiv preprint arXiv:2003.03485 (2020).
>
> Li, Zongyi, et al. "Multipole graph neural operator for parametric partial differential equations." Advances in Neural Information Processing Systems 33 (2020): 6755-6766.
>
> Lu, Lu, et al. "Learning nonlinear operators via DeepONet based on the universal approximation theorem of operators." Nature Machine Intelligence 3.3 (2021): 218-229.
>
> Zongyi Li, Nikola Borislavov Kovachki, Kamyar Azizzadenesheli, Kaushik Bhattacharya,
> Andrew Stuart, Anima Anandkumar, et al. Fourier Neural Operator for parametric partial
> differential equations. In International Conference on Learning Representations, 2020.
>
> These results have shown that operator based approaches not only give state-of-the-art accuracy, but are also not limited to predicting output values along a predetermined discretization, and are able to perform zero-shot super-resolution.  Given that the focus of our paper was about improving state-of-the art operator learning methods we did not feel it necessary to repeat this comparison which is already present in previous work.
>
> The reviewer is correct that in general, the accuracy of the model will depend on various hyperparameters and the available data.  However the lower bounds and discussion in Section 3 show that the limitations of linear decoders exist independent of hyperparameter settings and training data.  Therefore, the focus of the experiments was to show the performance increase from using a nonlinear decoder, even without additional hyperparameter tuning.  The reviewer also brings up an important point that the performance of such models will be affected by the resolution of measurements of input and output functions available.  When functions are less continuous or have larger variation, more measurement points will be needed to characterize their behavior.  Some theoretical results on the effect of the measurement resolutions for DeepONet and FNO models can be found in
>
> Lu, Lu, et al. "Learning nonlinear operators via DeepONet based on the universal approximation theorem of operators." Nature Machine Intelligence 3.3 (2021): 218-229.
>
> Kovachki, Nikola, Samuel Lanthaler, and Siddhartha Mishra. "On universal approximation and error bounds for Fourier Neural Operators." Journal of Machine Learning Research 22 (2021)
>
> Obtaining similar theoretical results for NOMAD is an interesting direction for future work.
>
> In response to the reviewer's questions:
>
> By computational efficiency we refer to the training time and the size of the model (in terms of number of parameters).  We give information on these quantities in Table 1 for the shallow water benchmark where there is a clear order of magnitude difference between NOMAD and the other methods for the number of parameters and training time.
>
> The probability measure mu defined in line 132 is the same probability measure mu that appears in equation (5).  It is a probability measure on the space of input functions, $L^2(\mathcal X)$.  While the quantity inside the expectation of (5) is a norm on the space of output functions, it is represented as a random variable that depends on the input function (through the candidate and ground truth operators). Therefore this is well defined as an expectation with respect to the probability measure mu.
>
> The set U that appears in (6) is a compact set of input functions.  We thank the reviewer for pointing out this was not defined earlier.  We have added a comment before (6) to address this.
>
> We thank the reviewer for pointing out the typo in line 260.

---

> > ### Comment · Reviewer_Dpah · 2022-08-09
> > **comment to response**
> >
> >
> > Dear colleagues,
> >
> > Thank you for your response. Partially it addressed my concerns. However, some issues still slightly outweigh the merits of this paper, so I can not increase a score to a full extent. In particular,
> >
> > - although the authors proposed a new insight on how we can represent a model of a nonlinear functional operator, still some ablation study is needed to understand to what extent the approach based on this model architecture is stable/robust, namely, how the hyperparameters influence the final results, how the dimensionality of the latent space influences the final results, how the capacity of the decoder influences the final accuracy, etc.
> >
> > - the idea of interpreting a linear decoder as a linear combination of some basis functions is helpful, although it is not novel, since for MLPs, many approaches tried to exploit such structure of the last linear layer when training the parameters. So the technical contribution of the paper is moderate to a significant extent,
> >
> > - the performance of the final models is affected by the resolution of measurements of input and output functions, so the authors did not exploit anyhow structure of input/output functions (e.g., smoothness, some spatial correlations) to make the approach mode robust,
> >
> > - concerning the comparison with a Koopman operator. I consider this as some other realization of a scheme encode->approximate->decode, in which the authors of corresponding papers also impose some sequential constraints on the approximation, and/or on the encoder, a and/or possibly, on the decoder.
> >
> > - I would propose to include more benchmarks in a follow-up paper.

---

### Official Review · Reviewer_AwfW · 2022-07-19

**Rating:** 7
**Confidence:** 2
**Soundness:** 3 good
**Presentation:** 4 excellent
**Contribution:** 3 good

**Summary:**

This paper proposes a supervised learning method called NOMAD to learn operators (maps between infinite-dimensional function spaces). Like many previous operator-learning methods, they have an encoder (from function space to R^m), an approximator (from R^m to R^n) approximating the operator, and a decoder (from R^n to the new function space). Unlike previous papers with this framework, the decoder can be nonlinear, with the idea that although there are guarantees about linear representations, nonlinear representations can be more efficient/lower-dimensional. They test their method on the antiderivative operator, linear transport & nonlinear wave propagation and compare NOMAD to state-of-the-art methods on the 3rd example. NOMAD has better or close accuracy, but with fewer latent dimensions and increased computational efficiency.

**Questions:**

FNO is "resolution-invariant." Is a limitation of NOMAD that the grid needs to be consistent across examples?

**Limitations:**

I thought that the limitations section was well-written and unusually thoughtful.

**Strengths And Weaknesses:**

*Originality:* To my understanding, NOMAD's architecture is not very different from some of the previous papers, but has a nonlinear decoder. However, the previous methods are presented in a unified way that is easy to understand, and the authors make a solid argument for these methods being consistently restricted by having a linear decoder.

*Quality:* I think that the claims are sound. The paper could be strengthened with more extensive experiments comparing against previous papers. It's also mentioned in the appendix that they compare against the simplest possible DeepONet so that they can isolate the effect of the decoder. Perhaps this is not a fair representation of the power of DeepONet?

*Clarity:* There is some tricky math in this paper, but overall, I think that the main points were clear. I appreciated the anti-derivative & parametric advection PDE examples for building intuition that there can be some low-dimensional structure that linear decoders cannot capture.

*Significance:* If NOMAD really is a more accurate and more efficient approach to operator learning than methods like DeepONet & FNO, which are getting a lot of attention, then it could be very impactful. As mentioned above, I think the argument could be strengthened with more experiments comparing the methods.



UPDATE: the response was helpful, and I'm increasing my rating. Thanks!

---

> ### Author Response · Authors · 2022-08-02
> **Response to Reviewer AwfW**
>
> We thank the reviewer for their time and feedback.
>
> There have been several interesting and effective modifications of DeepONet that can improve its performance, for example see
>
> Lu Lu, Xuhui Meng, Shengze Cai, Zhiping Mao, Somdatta Goswami, Zhongqiang Zhang,
> and George Em Karniadakis. A comprehensive and fair comparison of two neural operators
> (with practical extensions) based on FAIR data. Computer Methods in Applied Mechanics and
> Engineering, 393:114778, 2022
>
> P Clark Di Leoni, Lu Lu, Charles Meneveau, George Karniadakis, and Tamer A Zaki. Deep-
> onet prediction of linear instability waves in high-speed boundary layers. arXiv preprint
> arXiv:2105.08697, 2021
>
> However, all of the modifications still use a linear decoder and therefore are subject to the lower bound presented in the paper. The goal of the antiderivative and advection experiments is to show that even the simplest DeepONet architecture augmented with a nonlinear decoder (the implementation of NOMAD used in the paper) achieves state of the art performance without any of the additional modifications to DeepONet available.  For the free-surface waves experiment experiment we used a DeepONet with the harmonic feature expansion suggested in the above two references while keeping the same architecture for NOMAD; this has been clarified in the paper.  We expect that incorporating these elements into the implementation of NOMAD would improve performance as well, but the intent of the paper was to focus on the effect of the decoder.
>
> The resolution invariance of FNO in the input functions can be incorporated into NOMAD as well.  The implementation of NOMAD presented in the paper uses an encoder that indeed requires consistent input function measurement locations. This condition can be relaxed in the same manner as in
>
> Michael Prasthofer, Tim De Ryck, and Siddhartha Mishra. Variable-input deep operator networks.
> arXiv preprint arXiv:2205.11404, 2022
>
> by considering variable measurements for the input functions. However other encoder choices could be used which have the kind of resolution invariance that results from using the Fourier transform as in FNO.  For example, instead of the encoder taking point-wise measurements of the input function, it could project the input function onto the first elements of a chosen basis.  This is computed by integrating the input function against these basis functions, for which there are approximations that can accept varying resolutions and even non-uniform meshes.  If these basis functions are trigonometric polynomials, then this projection is the Fourier transform.
>
> For the output function, NOMAD has no restrictions on which points can be queried.

---

### Official Review · Reviewer_ixP9 · 2022-07-26

**Rating:** 8
**Confidence:** 5
**Soundness:** 4 excellent
**Presentation:** 4 excellent
**Contribution:** 3 good

**Summary:**

The authors contribute a thorough, sound, and clearly explained lower bound to the approximation quality of learned operators with linear decoders. Further, the authors propose NOMAD: an extension of DeepONet to nonlinear decoders which significantly outperforms DeepONet in terms of accuracy at a given number of model parameters on the shallow-water equation, advective flow, and integrating a cosine function.


**Questions:**

3.1 Is the universal operator approximation theorem valid for NOMAD? DeepONet has shown that a universal operator approximation theorem can be shown for a linear decoder. Replacing the linear decoder with an MLP seems to invalidate that theorem.

3.2 Can NOMAD exploit and learn spatial correlations? Similar to DeepONet, NOMAD uses the space, x, as input. So, there does not seem a step to learn spatial correlations, e.g., via CNNs, GNNs, Fourier transforms, or other.

3.3 Could NOMAD also be used to replace the linear decoder in FNO? Or does NOMAD only work in combination with DeepONet?


**Limitations:**

4.1 The authors clearly state that the efficiency of NOMAD relies on the existence of low-dimensional nonlinear solution submanifolds and clearly state the PDE cases for which this is might not be true.

4.2 It would be helpful for the future reader, if the authors address the mentioned questions in the camera ready version.

4.3 The paper includes a satisfactory paragraph on potential negative societal impact.


**Strengths And Weaknesses:**

1. Strengths:

1.1 The general topic area of learning operators is of very high significance for reduced-order modeling, uncertainty quantification, parameter exploration, and sensitivity analysis in climate science, physics, fluid dynamics and more. Existing architectures, such as DeepONet and FNO, have shown proof of concepts but can be improved in accuracy.

1.2 The authors convey the limitations of current operator learning architectures with linear decoders with great clarity and intuition. The limitations are supported by a sound proof on the lower bound on the approximation and quality references to relevant works in the reduced-ordor modeling literature. The math is sound and all variables are defined with name and dimensions.

1.3 The proposed method to replace a linear with nonlinear decoding is easy to implement.

1.4 The results support the claimed contributions. Indeed, NOMAD is more accurate than DeepONet if the latent space is limited to a few parameters.

2. Weaknesses:

2.1 The results contain three PDEs that have been selected to highlight the benefit of NOMAD. It would have been interesting to see how NOMAD performs on standard benchmarks that have been used in FNO, DeepONet paper; specifically incompressible  Navier-Stokes at low Reynolds number. It would have also been interesting to include FNO in Table 1. However, the results section is of sufficient quality and extent to support the authors claims.

2.2 NOMAD does not address how to do autoregressive prediction, but it also seems out of scope for the strong theoretical contributions that NOMAD makes.

---

> ### Author Response · Authors · 2022-08-02
> **Response to Reviewer ixP9**
>
> We thank the reviewer for their comments and evaluation.
>
> Regarding the first question, a universal approximation property should hold for NOMAD.  There are two ways to approach this.  The first is somewhat indirect and makes use of the universal approximation result for linear decoders.  This result guarantees the existence of a sufficiently high latent dimension and linear decoder such that the overall approximation error is within a desired tolerance.  If this linear decoder is replaced with an MLP, it would then suffice to show there exists an MLP which is able to approximate this linear decoder to arbitrary accuracy.  This follows from the universal approximation property of MLPs (with suitably chosen architecture and activations).
>
> An approach that does not rely on a reduction to the linear case could potentially work as follows.  The universal approximation theorems for architectures with linear decoders have an assumption that allows finite dimensional linear representations to give good accuracy in a particular norm.  For example, in the supremum norm compactness of the set of input functions and continuity of the operator assumptions tell us that the images of linear decoders can capture the output functions.  In the $L^2$ sense, it is the assumption that the probability measure associated with the output functions has a finite second moment.  A corresponding result for NOMAD would need an analogous assumption as the starting point.  The operator learning manifold hypothesis as stated in the paper would suffice for such an assumption in an $L^2$ sense, but there may be other options involving a low dimensional structure of the input functions and good continuity (Lipschitz) properties of the operator.  Then, it would remain to be shown that using an MLP as the decoder is expressive enough to have the target manifold of output functions contained in its image.  This is indeed an interesting direction for future work.
>
> The implementation of NOMAD presented in the paper intentionally used only simple feed-forward MLPs as its building blocks.  The purpose of this was to demonstrate the significant performance improvement of a nonlinear decoder in the absence of architectural components which explicitly can learn additional structure, such as spatial correlations.  The defining property of NOMAD is the presence of a nonlinear decoding map; as long as this is satisfied, NOMAD could have encoding, approximation, and decoding maps which do make use of neural network architectures that can learn or exploit spatial correlations in the data.  For example, if the input is measured on a regular grid, it could be processed with a convolutional network, a GNN, or projected onto a Fourier basis.  Similarly, instead of a simple MLP as the decoder with concatenated latent variables and query location, a transformer architecture or a hyper-network could be used to generate the output functions.
>
> To address the last question, the component of the FNO that uses an encoding/decoding architecture is the evaluation of the kernel integral transform via the Fourier convolution theorem.  The decoding component of this computation is the use of the inverse Fourier transform to reconstruct a function from its first $n$ Fourier coefficients.  It is possible to replace this step with a fully nonlinear decoder, however this would no longer result in a function corresponding to the evaluation of that kernel integral transform.

---

### Author Response · Authors · 2022-08-09
**Rebuttal Discussion Reminder**

We would like to thank all the reviewers again for their feedback.  Please let us know if there are any remaining questions or clarifications we can address.

---

### Meta-Review · Area_Chair_KauL · 2022-08-20

**Recommendation:** Accept
**Confidence:** Certain

**Metareview:**

The focus of the submission is supervised learning with functional inputs and outputs. Particularly, the authors consider the
encoder-approximator-decoder architecture (2)-(3) to tackle this task. After discussing the limitations of linear decoders in this scheme (meant in L^2 and uniform sense; the latter is elaborated in Proposition 1), the authors present the nonlinear NOMAD architecture under the assumption of operator learning manifold hypothesis (12) which captures a low-dimensional output space condition. They demonstrate the efficiency of the approach compared to the DeepONet method relying on linear decoders, Fourier neural operators and LOCA (i) when using stochastic gradient descent on the empirical loss (1), (ii) on the learning problem of the antiderivative operator and learning the solution of two partial differential equations.

Functional data analysis is a fundamental area of machine learning with a large number of applications. The authors present a new method in this context which can be of definite interest to the community as it was assessed by the reviewers.

**Award:**

No

---

### Decision · Program_Chairs · 2022-09-14

Accept